# An Efficient Method for Selective Syntheses of Sodium Selenide and Dialkyl Selenides

**DOI:** 10.3390/molecules27165224

**Published:** 2022-08-16

**Authors:** Na Hye Shin, Yoo Jin Lim, Chorong Kim, Ye Eun Kim, Yu Ra Jeong, Hyunsung Cho, Myung-Sook Park, Sang Hyup Lee

**Affiliations:** College of Pharmacy and Innovative Drug Center, Duksung Women’s University, 33, Samyangro 144-gil, Dobong-gu, Seoul 01369, Korea

**Keywords:** diorganyl selenide, sodium selenide, sodium borohydride, organoselenium, selenium reduction

## Abstract

The studies on the selective synthesis of dialkyl selenide compounds **1** were presented. Overcoming the complexity and difficulty of selenides (R-Se-R) and/or multiselenides (R-Se_n_-R; *n* ≥ 2), we aimed to optimize the reaction condition for the tolerable preparation of sodium selenide (Na_2_Se) by reducing Se with NaBH_4_, and then to achieve selective syntheses of dialkyl selenides **1** by subsequently treating the obtained sodium selenide with alkyl halides (RX). Consequently, various dialkyl selenides **1** were efficiently synthesized in good-to-moderate yields. The investigations on reaction pathways and solvent studies were also described.

## 1. Introduction

Selenium (Se), one of the essential trace elements, belongs to group 16 elements, also known as the chalcogen family [1]. During the last 40 years, organoselenium compounds have attracted attention as important reagents and intermediates in organic synthesis as well as in medicine [2,3]. Selenium compounds seem to play important roles in biological systems as antioxidant, chemopreventive, anti-inflammatory, and antiviral agents [4]. Selenium is used to form selenoamino acids, selenocystein (Sec) and selenomethionine (SeMet) that are incorporated as surrogates to sulfur derivatives [5,6]. In many respects, selenium and sulfur have very similar physical and chemical properties, but at the same time may display keen differences [7]. In general, the strengths of selenium–carbon bonds (234 kJ/mol) are weaker than those of sulfur–carbon bonds (272 kJ/mol), and the bond length of selenium–carbon (198 pm) is longer than that of sulfur–carbon (C-S, 184 pm) [7]. Selenium compounds are more nucleophilic and more acidic (pKa is 3.74 for H_2_Se) than sulfur compounds (pKa is 7.0 for H_2_S) [8]. Despite these intriguing characters, organoselenium compounds have not been extensively studied for decades, because of their irregular reactivity and instability, biological toxicity, and lack of tolerable synthetic methods. However, currently organoselenium compounds have been used in many applications, including synthetic methodologies and medicinal purposes for antioxidant [9,10] and anticancer agents [11,12,13,14].

Among many organoselenium compounds, selenides (R-Se-R) and diselenides (R-Se-Se-R) are the most traditional classes of organoselenium compounds. Nevertheless, studies and derivatization of selenide compounds are less reported in comparison to diselenide compounds. The incorporated form of Se in selenoamino acids such as SeMet is the dialkyl selenide and so, we focused on the selective synthesis of dialkyl selenides. So far, several methods have been reported for the synthesis of dialkyl selenides [15,16,17,18,19,20]. Among them, reactions of sodium selenide (Na_2_Se) generated in situ and an organic halide in an appropriate solvent have been employed to prepare a small number of symmetrical dialkyl selenides. In these methods, selenium is reduced to selenide anion(s) by proper reducing agents. For this purpose, NaBH_4_ [18], NaH [19], Na [20], and Li(CH_3_CH_2_)_3_BH [2] have been used with corresponding reaction conditions. Most of the methodologies, however, suffer from various limitations, such as poor reproducibility, narrow tolerance according to structures, poor yields and product instability, side product formation, and tricky manipulations. In reality, we experienced poor reproducibility of previously known procedures, and severe difficulty in manipulating products due mainly to their chemical instabilities. In particular, we observed that, along with desired selenides, diselenides and multiselenides were generally formed as side products, and dark solids were also formed in storing for even a short period of time.

According to the previous reports [16,18] and our preliminary studies, we found that NaBH_4_ might be an appropriate reducing agent for this transformation; thus, we focused on the use of NaBH_4_. We here report the studies on the selective formation of sodium selenide and corresponding dialkyl selenides in a one-pot reaction, along with the studies on reaction pathways and solvent effects.

## 2. Results and Discussion

### 2.1. Optimization for the Reaction Conditions

In order to achieve selective syntheses of dialkyl selenides (R-Se-R), we first aimed to investigate the reaction conditions for the selective formation of sodium selenide (Na_2_Se). Based on previous procedures for selenides [18], we attempted to apply these reaction conditions for various aliphatic selenides with proper modifications. For this purpose, we chose to employ selenium (Se) and NaBH_4_ as a reducing agent for the formation of sodium selenide. The sodium selenide generated in situ was treated with proper electrophiles such as alkyl halide (RX) to give desired dialkyl selenides **1**. So, we conducted the reactions with benzyl bromide (BnBr) as a template alkyl halide, leading to the selective synthesis of dibenzyl selenide (**1a**), as shown in Figure 1.

We performed the reactions for the synthesis of sodium selenide under the condition of Se and NaBH_4_ in THF-H_2_O solvent. On the basis of the importance of reducing agent in these reactions, we optimized the reaction conditions by varying the amounts of NaBH_4_ and the reaction time at room temperature (25 °C). Then, the resulting sodium selenide was directly reacted with BnBr to give the product **1a**. Regarding the amount of BnBr, we intended to provide a sufficient amount of it. Thus, we tested BnBr in the range of 2.0–3.0 eq, and found that 2.4 eq of BnBr is an adequate amount, unless otherwise mentioned (data not shown). As shown in Table 1, first we investigated the reaction conditions for the formation of sodium selenide. Based on our preliminary studies and the information that the theoretical amount of NaBH_4_ is 2.0 eq (vs. Se), we chose the range of 1.0–3.0 eq for NaBH_4_, and 25 °C for the reaction temperature. As expected, the amount of NaBH_4_ was found to be the most important factor, and 3.0 eq of this reagent gave the best yield in the given range. When we used a lower amount (e.g., 1.0 eq) of that, we observed the substantial formation of side product, dibenzyl diselenide **2a**. As the amount of NaBH_4_ was increased, the formation of **2a** was decreased, which is consistent with the suggested reaction mechanism (see below) and our previous observation [21]. Considering the required stoichiometric amount (2.0 eq) of the reducing agent, the use of 3.0 eq seemed to be excessive, but we believed that the excess amount of reducing agent could be required due to the presence of oxygen, incomplete workings of the reducing agent, and/or extraconsumption in cleaving side products **2**, diselenides (Figure 1). In addition, the reaction time for the formation of sodium selenide could affect the final results. Interestingly, as the reaction time is elongated, the product **1a** (selenide) formation is decreased with the increased formation of side product **2a** (diselenide). It is believed that when the reaction time for sodium selenide is getting longer, the sodium selenide could gradually oxidize to sodium diselenide, leading to the higher generation of **2a**. Taken together, the best condition for the two-step reaction was: (1) Se 1.0 eq, NaBH_4_ 3.0 eq and 1 h reaction time; (2) BnBr 2.4 eq and 2 h reaction time at room temperature.

### 2.2. Reaction Pathways

According to the results so far, we wish to perform mechanistic studies on reaction pathways. Here, we propose reaction pathways for the formation of dialkyl selenides **1** and dialkyl diselenides **2**, as shown in Figure 2. At first, elemental Se could be reduced by NaBH_4_ to diselenide dianion (Se_2_^2−^, oxidation state: —1), which could also be further reduced to selenide dianion (Se^2−^, oxidation state: —2). The two species (Se_2_^2−^ and Se^2−^) could equilibrate each other according to redox condition. Then, selenide dianion (Se^2−^) could react with alkyl halide to give the products, selenides **1**, and similarly, diselenide dianion (Se_2_^2−^), to give side products, diselenides **2**. Notably, we could presume and indirectly verify the presence of the selenium species by capturing them with alkyl halide. As expected, the side products **2** were continuously formed in most of the reactions, and the control of selective synthesis of selenides **1** over diselenides **2** was a challenge. Therefore, we intended to employ sufficient amount of reducing agent to avoid the formation of diselenide dianion and dialkyl diselenides **2**. Once **2** is formed, it could also undergo further reduction to give selenol **3**, with further consumption of reducing agent. Taken together, we suggested the pathways for compounds **1**–**3** under reduction condition (e.g., NaBH_4_).

### 2.3. Synthesis of Dialkyl Selenides **1**

Using this optimized condition, we achieved the selective syntheses of various dialkyl selenides **1**, expanding the scope of this reaction by applying different aliphatic halides, as shown in Table 2. We tried to employ primary, secondary, and tertiary alkyl bromides. At first, benzyl bromide, phenethyl bromide, and allyl bromide afforded good yields (Table 2, entries 1 to 3). As primary aliphatic bromides, *n*-butyl, *n*-pentyl, *n*-hexyl, and *n*-octyl bromide also gave good yields (entries 4 to 7). As secondary alkyl halides, *c*-pentyl, *c*-hexyl, 3-pentyl, and 4-heptyl bromide provided moderate-to-poor yields (entries 8 to 11). However, tertiary halides (e.g., *t*-butyl, *t*-amyl, and trityl bromide) did not afford an appreciable amount of product, due mainly to steric hindrance (data not shown). In addition, aromatic bromides generally gave poor results. Phenyl bromide, phenyl bromides with an electron-donating substituent (e.g., methyl group), and phenyl bromides with electron-withdrawing substituents (e.g., cyano or trifluoromethyl group) afforded poor yields (5–10%, data not shown) even at a high reaction temperature (e.g., 153 °C). In order to improve the results, we also tested the additional use of sodium iodide as a catalyst or tetrabutylammonium bromide as a phase-transfer catalyst (PTC), leading to no appreciable success. We also attempted to employ aromatic iodides instead of aromatic bromides under harsh reaction conditions (e.g., elongated reaction time at 120 °C), without appreciable improvement. Notably, we were also interested in the formation of cyclic selenides and so we tested the use of alkyl dibromides (e.g., α,α′-dibromoxylene and 1,4-dibromo-2,3-dihydroxybutane, entries 12 and 13, respectively). For these reactions, we applied half amount of alkyl dibromides and a low concentration of reagents (e.g., 64 mM for Se) in order to facilitate intramolecular cyclization reactions. As expected, these reactions worked, leading to the generation of cyclic selenides in modest yields (65 and 51%). Notably, for compound **1m**, the used starting material ((±)-1,4–dibromo–2,3–butanediol) was an enantiomer mixture (two enantiomers (R,R)- and (S,S)-form) and thus the product **1m** was also an enantiomer mixture ((R,R)- and (S,S)-form).

### 2.4. Solvent Studies

We performed the reactions under THF–H_2_O as a co-solvent system and the results were generally successful. Nevertheless, we conducted solvent studies to find out better and more convenient conditions. So, we chose ethanol (EtOH) as a volatile protic solvent and acetonitrile (MeCN), tetrahydrofuran (THF), and dimethoxyethane (DME) as volatile aprotic solvents. Using these solvent systems, we performed the reactions under similar conditions by using NaBH_4_ and BnBr as alkyl halide at room temperature (25 °C), which was summarized in Table 3. The reaction for the preparation of Na_2_Se under EtOH solvent seemed to be in a white suspension condition and selectively provided dibenzyl selenide **1a** in a good yield (81%). The reactions under MeCN, THF, and DME solvents seemed to be in brown-to-reddish-brown suspension conditions with heterogeneous reaction media. The reaction under MeCN afforded dibenzyl selenide **1a** in a good yield (80%) with a trace amount of dibenzyl diselenide. However, the other reactions under THF and DME solvents required longer reaction times (48 and 47 h, respectively) and provided modest yields (50 and 30%, respectively). Although the reaction under EtOH solvent gave a good result (81%), the reproducibility of the reaction was not so good, probably due to the undesired participation of EtOH as a nucleophile producing benzylethyl ether (BnOEt). As a result, MeCN was found to be a good choice as a non-aqueous, volatile, aprotic solvent.

Furthermore, inspired by the good results under MeCN solvent, we expanded test reactions under MeCN using several alkyl halides. We also compared the reactions of alkyl bromides with those of alkyl chlorides. In general, the results under MeCN solvent were similar or lower than those under the original solvent (H_2_O-THF), and sometimes the results fluctuated according to the conditions, as shown in Table 4. Although the results under MeCN were found to be variable, it seemed a milestone experiment that sodium selenide and then dialkyl selenides **1** could be obtained in good yield in MeCN as a non-aqueous, volatile, aprotic solvent. Nevertheless, the heterogeneity issue might still limit broad applications, and further studies for improvements are in progress.

## 3. Experimental

### 3.1. General

Most of the reagents were purchased at good commercial quality and used without further purification, unless otherwise noted. Reactions were continuously monitored by thin-layer chromatography (TLC) using glass plates (0.25 mm) coated with silica gel (20 × 20 cm^2^; Aldrich No. Z12272-6) and visualized by UV light and staining solutions such as cerium molybdate and phosphomolybdic acid, if necessary. Preparative TLC (PTLC) and column chromatography were mainly employed for purifications. PTLC separations were carried out on the same silica gel plates for TLC. Column chromatography was conducted using Merck silica gels (230–400 mesh). Measurement of melting points was carried out in open capillary tubes using Electrothermal IA9100 melting point apparatus, and the measurements were uncorrected. FT-IR spectra were obtained on a Thermo Fisher (Waltham, MA, USA) FT-IR spectrometer (Nicolet Summit) and frequencies (ν) are given in reciprocal centimeters (cm^−1^). ^1^H (300 MHz) and ^13^C (75 MHz) NMR spectra were obtained by a Bruker (Billerica, Massachusetts, USA) DRX 300 spectrometer and the chemical shift values (δ) are expressed as units relative to tetramethylsilane (TMS). Mass spectra were obtained using EI or ESI ionization method. HPLC analyses were conducted using the following Waters Associate Units: 1525 Binary HPLC Pump, 2998 Photodiode Array Detector, and C_18_ COSMOSIL column (4.6 × 250 mm^2^). The product analyses were performed using a linear gradient condition. Condition 1: from 70% A (H_2_O) and 30% B (MeCN) for 3 min (isocratic) to 10% A and 90% B in 30 min (gradient), then to 10% A and 90% B for 7 min (isocratic) and, finally, from 10% A and 90% B to 70% A and 30% B over 3 min (gradient), then to 70% A and 30% B for 5 min (isocratic). Condition 2: from 50% A (H_2_O) and 50% B (MeCN) for 3 min (isocratic) to 10% A and 90% B in 30 min (gradient), then to 10% A and 90% B for 7 min (isocratic) and, finally, from 10% A and 90% B to 50% A and 50% B over 3 min (gradient), then to 50% A and 50% B for 5 min (isocratic). The flow rate was 1 mL/min, and the eluent was monitored at 254 nm. The HPLC solvents were filtered (aqueous solution with Millipore HVLP, 0.45 mm; acetonitrile with Millipore HV, 0.45 mm) and degassed before utilization.

### 3.2. General Procedure for the Synthesis of Dialkyl Monoselenides **1**

To a stirred mixture of NaBH_4_ (144 mg, 3.9 mmol, 3.0 eq) and H_2_O (2 mL), Se (100 mg, 1.3 mmol, 1.0 eq) was added under the nitrogen atmosphere. The resulting mixture was stirred for 1 h at room temperature turning white. Then, alkyl halides (3.1 mmol, 2.4 eq) in THF (8 mL) were slowly added and stirring was continued for 2–48 h at room temperature to 50 °C until completion of the reaction (Table 2). The reaction mixture was diluted with water (30 mL), extracted with CH_2_Cl_2_ (3 × 30 mL) and then washed with brine (30 mL). The combined organic layers were dried (anhydrous MgSO_4_), and concentrated in vacuo to give sticky residue. The residue was purified by column chromatography (1:50 → 1:1 EtOAc/*n*-hexanes) to give **1a**–**1m** as a light yellow oil, unless otherwise noted. The NMR charts for all compounds **1** were provided in Appendix A.

#### 3.2.1. 1,2-Dibenzyl Selenide (**1a**) [2]

Use of benzyl bromide (0.36 mL, 3.1 mmol, 2.4 eq), 2 h reaction time at room temperature in general procedure, afforded the title compound **1a** (310 mg, 87%) as a light yellow solid. Mp 44–45 °C; *R*_f_ 0.34 (1:20 EtOAc/*n*-hexanes); HPLC t*_R_* 31.17 min (condition 1); λ_max_ = 215 nm; IR (KBr) 3033, 3030, 703, 696, 670, 667 cm^−1^; ^1^H-NMR (300 MHz, CDCl_3_) *δ* 7.32–7.18 (m, 10H, Ar), 3.72 (s, 4H, CH_2_); ^13^C-NMR (75 MHz, CDCl_3_) *δ* 139.37 (Ar), 129.17 (Ar), 128.69 (Ar), 27.75 (CH_2_); MS *m/z* 262 [M]^+^; HRMS (+ESI) calcd for C_14_H_14_Se [M]^+^ 262.0261, found 262.0264.

#### 3.2.2. 1,2-Bis(2-phenylethyl) Selenide (**1b**) [2]

Use of phenethyl bromide (0.42 mL, 3.1 mmol, 2.4 eq), 5 h reaction time at room temperature, in general procedure afforded the title compound **1b** (248 mg, 66%) as a light yellow solid. Mp 148–156 °C; *R*_f_ 0.30 (1:20 EtOAc/*n*-hexanes); HPLC t*_R_* 30.54 min (condition 1); λ_max_ = 261 nm; IR (KBr) 3104, 1029, 705 cm^−1^; ^1^H-NMR (300 MHz, CDCl_3_) *δ* 7.36–7.10 (m, 10H, Ar), 2.95 (t, *J* = 7.8 Hz, 4H, SeCH_2_), 2.78 (t, *J* = 7.8 Hz, 4H, PhCH_2_); ^13^C-NMR (75 MHz, CDCl_3_) *δ* 141.42 (Ar), 128.66 (Ar), 128.55 (Ar), 126.52 (Ar), 37.36 (SeCH_2_), 25.19 (PhCH_2_); MS *m/z* 290 [M]^+^; HRMS (+ESI) calcd for C_16_H_18_Se [M]^+^ 290.0574, found 290.0570.

#### 3.2.3. 1,2-Diallyl Selenide (**1c**) [22]

Use of allyl bromide (0.26 mL, 3.1 mmol, 2.4 eq), 25 h reaction time at room temperature, in general procedure afforded the title compound **1c** (135 mg, 66%). Bp 56–58 °C; *R*_f_ 0.51 (*n*-hexane); HPLC t*_R_* 33.86 min; λ_max_ = 243 nm (condition 1); IR (KBr) 987, 739 cm^−1^; ^1^H-NMR (300 MHz, CDCl_3_) *δ* 5.89–5.80 (m, 2H, CH_2_=CH), 5.05–5.00 (m, 4H, CH_2_=CH), 3.15 (d, *J* = 7.6 Hz, 4H, SeCH_2_); ^13^C-NMR (75 MHz, CDCl_3_) *δ* 135.19 (CH_2_=*C*H), 116.61 (*C*H_2_=CH), 25.55 (SeCH_2_); MS *m/z* 162 [M]^+^.

#### 3.2.4. 1,2-Di-*n*-butyl Selenide (**1d**) [19]

Use of *n*-butyl bromide (0.33 mL, 3.1 mmol, 2.4 eq), 3 h reaction time at room temperature, in general procedure afforded the title compound **1d** (228 mg, 93%). Bp 72–74 °C; *R*_f_ 0.49 (1:50 EtOAc/*n*-hexanes); HPLC t*_R_* 32.73 min (condition 1); λ_max_ = 222 nm; IR (KBr) 2953, 2872, 994, 741 cm^−1^; ^1^H-NMR (300 MHz, CDCl_3_) *δ* 2.56 (t, *J* = 7.5 Hz, 4H, SeCH_2_), 1.64 (quintet, *J* = 7.4 Hz, 4H, SeCH_2_CH_2_), 1.40 (sextet, *J* = 7.4 Hz, 4H, CH_2_CH_3_), 0.92 (t, *J* = 7.3 Hz, 6H, CH_3_); ^13^C-NMR (75 MHz, CDCl_3_) *δ* 32.99 (SeCH_2_), 23.84 (SeCH_2_*C*H_2_), 23.29 (*C*H_2_CH_3_), 13.83 (CH_3_); MS *m/z* 194 [M]^+^; HRMS (+ESI) calcd for C_8_H_18_Se [M]^+^ 194.0574, found 194.0575.

#### 3.2.5. 1,2-Di-*n*-pentyl Selenide (**1e**) [2]

Use of *n*-pentyl bromide (0.38 mL, 3.1 mmol, 2.4 eq), 3 h reaction time at room temperature, in general procedure afforded the title compound **1e** (230 mg, 82%). Bp 96–106 °C; *R*_f_ 0.50 (*n*-hexanes); HPLC t*_R_* 35.94 min (condition 2); λ_max_ = 224 nm; IR (KBr) 2957, 2925, 1466, 1241, 728 cm^−1^; ^1^H-NMR (300 MHz, CDCl_3_) *δ* 2.55 (t, *J* = 7.5 Hz, 4H, SeCH_2_), 1.66 (quintet, *J* = 7.3 Hz, 4H, SeCH_2_CH_2_), 1.36–1.34 (m, 8H, (CH_2_)_2_CH_3_), 0.90 (t, *J* = 6.9 Hz, 6H, CH_3_); ^13^C-NMR (75 MHz, CDCl_3_) *δ* 32.39 (SeCH_2_), 30.59 (SeCH_2_*C*H_2_), 24.16 (*C*H_2_CH_2_CH_3_), 22.45 (*C*H_2_CH_3_), 14.18 (CH_3_); MS *m/z* 222 [M]^+^; HRMS (+ESI) calcd for C_10_H_22_Se [M]^+^ 222.0887, found 222.0889.

#### 3.2.6. 1,2-Di-*n*-hexyl Selenide (**1f**) [19]

Use of *n*-hexyl bromide (0.43 mL, 3.1 mmol, 2.4 eq), 5 h reaction time at room temperature, in general procedure afforded the title compound **1f** (269 mg, 85%). Bp 90–94 °C; *R*_f_ 0.50 (*n*-hexanes); HPLC t*_R_* 26.92 min (condition 2); λ_max_ = 271 nm; IR (KBr) 3409, 2856, 1458, 743, 698 cm^−1^; ^1^H-NMR (300 MHz, CDCl_3_) δ 2.55 (t, *J* = 7.5 Hz, 4H, SeCH_2_), 1.65 (quintet, *J* = 7.4 Hz, 4H, SeCH_2_CH_2_), 1.43–1.28 (m, 12H, (CH_2_)_3_CH_3_), 0.89 (t, *J* = 6.8 Hz, 6H, CH_3_); ^13^C-NMR (75 MHz, CDCl_3_) *δ* 31.59 (SeCH_2_), 30.87 (SeCH_2_*C*H_2_), 29.90 (Se(CH_2_)_2_*C*H_2_), 24.20 (*C*H_2_CH_2_CH_3_), 22.78 (*C*H_2_CH_3_), 14.26 (CH_3_); MS *m/z* 250 [M]^+^; HRMS (+ESI) calcd for C_12_H_26_Se [M]^+^ 250.1200, found 250.1203_._

#### 3.2.7. 1,2-Di-*n*-octyl Selenide (**1g**) [23]

Use of *n*-octyl bromide (0.53 mL, 3.1 mmol, 2.4 eq), 5 h reaction time at room temperature, in general procedure afforded the title compound **1g** (330 mg, 85%). Bp 132–148 °C; *R*_f_ 0.60 (1:100 EtOAc/*n*-hexanes); HPLC t*_R_* 28.82 min (condition 1); λ_max_ = 297 nm; IR (KBr) 3418, 2914, 750, 696 cm^−1^; ^1^H-NMR (300 MHz, CDCl_3_) δ 2.55 (t, *J* = 7.5 Hz, 4H, SeCH_2_), 1.65 (quintet, *J* = 7.4 Hz, 4H, SeCH_2_CH_2_), 1.36–1.27 (m, 20H, (CH_2_)_5_CH_3_), 0.88 (t, *J* = 6.7 Hz, 6H, CH_3_); ^13^C-NMR (75 MHz, CDCl_3_) *δ* 32.05 (SeCH_2_), 30.90 (SeCH_2_*C*H_2_), 30.22 (Se(CH_2_)_2_*C*H_2_), 29.42 (Se(CH_2_)_3_*C*H_2_), 29.36 (*C*H_2_(CH_2_)_2_CH_3_), 24.19 (*C*H_2_CH_2_CH_3_), 22.88 (*C*H_2_CH_3_), 14.33 (CH_3_); MS *m/z* 306 [M]^+^; HRMS (+ESI) calcd for C_16_H_34_Se [M]^+^ 306.1826, found 306.1821.

#### 3.2.8. 1,2-Di-*c*-pentyl Selenide (**1h**) [24]

Use of *c*-pentyl bromide (0.31 mL, 3.1 mmol, 2.4 eq), 3 h reaction time at 50 °C, in general procedure afforded the title compound **1h** (138 mg, 50%). Bp 90–96 °C; *R*_f_ 0.47 (*n*-hexanes); HPLC t*_R_* 34.36 min (condition 1); λ_max_ = 225; IR (KBr) 2956, 2866, 1448, 1217, 932 cm^−1^; ^1^H-NMR (300 MHz, CDCl_3_) *δ* 3.25 (quintet, *J* = 7.1 Hz, 2H, SeCH), 2.12–2.01 (m, 4H, SeCH(CHH)_2_), 1.80–1.51 (m, 12H, SeCH(CHH*)*_2_(CH_2_)_2_); ^13^C-NMR (75 MHz, CDCl_3_) *δ* 37.76 (SeCH), 35.06 (SeCH(*C*H_2_)_2_), 25.19 (SeCH(CH_2_)_2_(CH_2_)_2_); MS *m/z* 218 [M]^+^; HRMS (+ESI) calcd for C_10_H_18_Se [M]^+^ 218.0574, found 218.0576.

#### 3.2.9. 1,2-Di-*c*-hexyl Selenide (**1i**) [25]

Use of *c*-hexyl bromide (0.38 mL, 3.1 mmol, 2.4 eq), 8 h reaction time at 50 °C, in general procedure afforded the title compound **1i** (150 mg, 48%). Bp 104–106 °C; *R*_f_ 0.50 (*n*-hexanes); HPLC t*_R_* 35.65 min (condition 1); λ_max_ = 228 nm; IR (KBr) 3422, 2849, 1724, 1447, 1258, 993, 699 cm^−1^; ^1^H-NMR (300 MHz, CDCl_3_) *δ* 2.96 (m, 2H, SeCH), 2.03–1.25 (m, 20H, CH_2_); ^13^C-NMR (75 MHz, CDCl_3_) *δ* 37.97 (SeCH), 35.40 (SeCH(CH_2_)_2_), 27.17 (SeCH(CH_2_)_2_(*C*H_2_)_2_), 26.08 (SeCH(CH_2_)_4_(CH_2_)_2_); MS *m/z* 246 [M]^+^; HRMS (+ESI) calcd for C_12_H_22_Se [M]^+^ 246.0887, found 246.0885.

#### 3.2.10. 1,2-Bis(3-pentyl) Selenide (**1j**) [22]

Use of 3-pentyl bromide (0.38 mL, 3.1 mmol, 2.4 eq), 5 h reaction time at 50 °C, in general procedure afforded the title compound **1j** (87 mg, 31%). Bp 90–100 °C; *R*_f_ 0.50 (*n*–hexanes); HPLC t*_R_* 34.60 min (condition 1); λ_max_ = 228 nm; IR (KBr) 2963, 2932, 1457, 1378, 1185, 861 cm^−1^; ^1^H-NMR (300 MHz, CDCl_3_) *δ* 2.69 (quintet, *J* = 6.3 Hz, 2H, CH), 1.75–1.60 (m, 8H, CH_2_), 0.99 (t, *J* = 7.3 Hz, 12H, CH_3_); ^13^C-NMR (75 MHz, CDCl_3_) *δ* 45.27 (CH), 28.63 (CH_2_), 12.29 (CH_3_); MS *m/z* 222 [M]^+^; HRMS (+ESI) calcd for C_10_H_22_Se [M]^+^ 222.0887, found 222.0883.

#### 3.2.11. 1,2-Bis(4-heptyl) Selenide (**1k**)

Use of 4-heptyl bromide (0.48 mL, 3.1 mmol, 2.4 eq), 25 h reaction time at 50 °C, in general procedure afforded the title compound **1k** (123 mg, 35%). Bp 118–120 °C; *R*_f_ 0.39 (*n*-hexanes); HPLC t*_R_* 32.85 min (condition 2); λ_max_ = 250 nm; IR (KBr) 3398, 2932, 2872, 1458, 1378, 756, 698 cm^−1^; ^1^H-NMR (300 MHz, CDCl_3_) *δ* 2.81 (quintet, *J* = 6.4 Hz, 2H, CH), 1.65–1.36 (m, 16H, CH_2_), 0.91 (t, *J* = 7.2 Hz, 12H, CH_3_); ^13^C-NMR (75 MHz, CDCl_3_) *δ* 41.21 (CH), 38.83 (CH*C*H_2_), 21.00 (CH_2_CH_3_), 14.26 (CH_3_); MS *m/z* 278 [M]^+^; HRMS (+ESI) calcd for C_14_H_30_Se [M]^+^ 278.1513, found 278.1514.

#### 3.2.12. 1,3-Dihydrobenzo[c]Selenophene (**1l**) [26]

Use of α,α′-dibromo-*o*-xylene (335 mg, 1.3 mmol, 1.0 eq), 24 h reaction time at room temperature, in general procedure afforded the title compound **1l** (151 mg, 65%) as a light yellow solid. Mp 36–37 °C; *R*_f_ 0.39 (*n*-hexanes); HPLC t*_R_* 22.40 min (condition 1); λ_max_ = 268 nm; IR (KBr) 3441, 3018, 1645, 1448, 834, 744 cm^−1^; ^1^H-NMR (300 MHz, CDCl_3_) *δ* 7.25–7.14 (m, 4H, Ar), 4.32 (s, 4H, CH_2_); ^13^C-NMR (75 MHz, CDCl_3_) *δ* 141.82 (Ar), 126.62 (Ar), 126.10 (Ar), 30.04 (CH_2_); MS *m/z* 184 [M]^+^; HRMS (+ESI) calcd for C_8_H_8_Se [M]^+^ 183.9791, found 183.9794.

#### 3.2.13. Tetrahydro-3,4-Selenophenediol (**1m**) [27]

Use of 1.4-dibromo-2,3-butanediol (315 mg, 1.3 mmol, 1.0 eq), 48 h reaction time at room temperature, and then extraction with *n*-butanol in general procedure afforded the title compound **1m** (108 mg, 51%) as a light yellow solid. Mp 77–80 °C; *R*_f_ 0.20 (3:1 EtOAc/*n*-hexanes); HPLC t*_R_* 28.12 min (condition 1); λ_max_ = 297 nm; IR (KBr) 3378, 1637, 1492, 1015, 704 cm^−1^; ^1^H-NMR (300 MHz, MeOH-d_4_) *δ* 4.20 (m, 2H, CH), 3.09, 2.79 (each dd, *J* = 10.2 and 9.6 Hz, each 2H, CH_2_); ^13^C-NMR (75 MHz, MeOH-d_4_) *δ* 80.02 (CH), 27.90 (CH_2_); MS *m/z* 168 [M]^+^; HRMS (+ESI) calcd for C_4_H_8_O_2_Se [M]^+^ 167.9690, found 167.9691.

## 4. Conclusions

We reported the systematic studies on the synthesis of Na_2_Se and corresponding dialkyl selenide compounds **1**. We established an optimized condition for these reactions as follows: (1) Se (1.3 mmol, 1.0 eq), NaBH_4_ (3.0 eq) in H_2_O for 1 h at room temperature; (2) alkyl bromide for 2–48 h with THF at room temperature to 50 °C. Applying this optimized condition using various alkyl halides, the desired dialkyl selenides **1** were successfully produced in good-to-moderate yields (31–93%). In particular, alkyl dihalide compounds were also employed to provide cyclic selenides, **1l** and **1m**, in 65% and 51% yields, respectively. Furthermore, solvent studies revealed that non-aqueous and volatile solvents (e.g., MeCN) other than THF-H_2_O also worked for these reactions. Taken together, we have completed systematic studies on the selective syntheses of Na_2_Se and the corresponding dialkyl selenide compounds **1** using NaBH_4_ as a reducing agent.

## Data Availability

The data presented in this study are available in insert article or Appendix A here.

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
