# Peer review of "An Efficient Method for Selective Syntheses of Sodium Selenide and Dialkyl Selenides"

_molecules, 2022, doi:10.3390/molecules27165224_

Round 1
Reviewer 1 Report
Lee and coworkers have presented interesting studies on selective synthesis of dialkyl selenide compounds, overcoming the complexity and difficulty of forming selenides (R-Se-R) and/or multiselenides (R-Sen-R; n≥2). The authors optimized the reaction conditions of sodium selenide (Na2Se) by reducing Se with NaBH4, and aimed to achieve selective syntheses of dialkyl selenides, forming various dilkyl selenides in good to moderate yields. The manuscript was carefully prepared with good characterization of and the work was unique in terms of selenium chemistry. The findings particularly on selenium-based THF should be highly important for preparation of related compounds. I thus highly recommend this paper and publication is recommended.
Reviewer 2 Report
In general, this work is based on known approaches, but it is of practical importance. The optimized conditions for the efficient preparation of organic selenides have been found and the experimental procedures have been described. The obtained results are useful for chemists, who work with elemental selenium and use selenium-centered nucleophiles in various reactions. Although this work is generally based on the known approaches, I recommend it for publication due to practical (experimental) importance. Some remarks:
Table 1 has the word "remarks" which I don't think is relevant at this point.
There is a phrase on the page 2: "According to our preliminary studies, we found that NaBH4 might be an appropriate reducing agents [16,18] for this transformation" . After reading this phrase, I got the impression that these ref. [16,18] refer to the authors' articles. However, it is not.
Compound 1m can exist as two diastereomers. If you can specify one or two diastereomers do you have, please, indicate it in the paper.
Reviewer 3 Report
In the manuscript entitled as “An Efficient Method for Selective Syntheses of Sodium Selenide and Dialkyl Selenides” the authors describe the synthesis of several dialkyl selenides obtained in low to good yields by the reaction between sodium selenide with alkyl halides. The sodium selenide was generated in situ by reducing Se with NaBH4. However, similar protocols for the synthesis of dialkyl selenides have been reported previously (such as Synlett 2004, 10, 1751-1754). Thus, despite the advances presented by the developed method, I suppose that this work is not suitable for publication in Molecules owing to the lack of novelty of the methodology presented.
In addition, the manuscript contains many incoherent points that need to be revised. In this way, I suggest that the authors consider some issues pointed out below for a future scientific publication:
· The introduction presents good arguments, however, the references used are not current. I suggest that the authors replace the references used by more current ones, when possible.
· Also, could present a schematic representation of previous work involving the synthesis of dialkyl selenides and their present work. This will help the readers to quickly review the state-of-the-art and present development.
· According to the optimization of the reaction conditions, the solvent mixture (THF and H2O) proved to be more efficient. Then, the authors perform another optimization involving the use of other solvents. It is unclear why this study was performed. Furthermore, the use of acetonitrile was not more efficient than the use of THF/H2O. Why was a comparative study done involving the use of different solvents in table 4?
· In section 2.3 on page 4, line 131, the authors described "When the alkyl bromides did not work sufficiently, the alkyl iodides were also tested". However, no results on the use of alkyl iodides are shown in Table 2. In which cases were alkyl iodides used? I suggest the authors add this information to table 2.
· Authors should review the 1H and 13C NMR spectra, for example, the chemical shifts in the spectra should be referenced according to the tetramethylsilane (TMS, δ 0.0 ppm) as the internal reference for 1H NMR and the solvent peak of CDCl3 (δ 77.23 ppm) for 13C NMR. In addition, the number of hydrogens observed in the 1H NMR spectrums must correspond to the total number of hydrogens of each compound. For example, for compound 1a the total number of hydrogens is 14. However, the value of the integrals in the spectrum is much smaller than expected. I suggest that authors review all spectra.
Round 2
Reviewer 3 Report
In the manuscript entitled as “An Efficient Method for Selective Syntheses of Sodium Selenide and Dialkyl Selenides” the authors describe the synthesis of several dialkyl selenides obtained in low to good yields by the reaction between sodium selenide with alkyl halides. The sodium selenide was generated in situ by reducing Se with NaBH4. Considering the changes performed by the authors, I observed an improvement in the quality of the manuscript. Therefore, I consider this article accepted for publication in Molecules.